# Bond Structures between Wood Components and Citric Acid in Wood-Based Molding

**DOI:** 10.3390/polym13010058

**Published:** 2020-12-25

**Authors:** Daisuke Ando, Kenji Umemura

**Affiliations:** Research Institute for Sustainable Humanosphere, Kyoto University, Gokasho, Uji, Kyoto 611-0011, Japan; umemura@rish.kyoto-u.ac.jp

**Keywords:** chemical reaction mechanism, citric acid, natural wood adhesive, wood-based molding

## Abstract

Citric acid-based wood adhesive is considered a chemical-bonding wood adhesive. However, the detailed structures of the bonds between wood components and citric acid remain unknown. Here, we examine the chemical bonding structures between citric acid and wood by heteronuclear single quantum coherence-nuclear magnetic resonance (HSQC-NMR) analysis of wood-based molding using Japanese cedar (*Cryptomeria japonica*) and citric acid. In the HSQC-NMR spectrum of the wood molding, some esterified C/H correlation peaks appeared. The primary hydroxyl groups of polysaccharides, such as cellulose and galactoglucomannan, and the primary hydroxyl groups of the β-O-4 and β-5 substructures in lignin were found to be esterified with citric acid. In contrast, the secondary hydroxyl groups, except for xylan, barely reacted because of the steric hindrance. Additionally, the C/H correlation peak volumes of the reducing ends of mannan and xylan in the anomeric region increased after molding. It was clarified that the glycosidic bonds in the hemicelluloses were cleaved under the acidic molding condition with citric acid. The HSQC-NMR analysis revealed that the esterification of hemicellulose and lignin, and degradation of hemicellulose, proceeded under the molding condition. These results will promote understanding of the adhesive mechanism of citric acid-based wood adhesive and of the properties of the molding.

## 1. Introduction

Wood adhesives have long been used for the efficient utilization of lignocellulosic resources, and many of these adhesives, are derived from petrochemicals [1,2]. In particular, formaldehyde-based wood adhesives, such as phenol-formaldehyde [3,4], urea-formaldehyde [5], and melamine-formaldehyde [6], account for a large portion of wood adhesives. However, our society is facing a shift away from fossil fuels and toward new more sustainable energy sources, with the ultimate goal of resolving the many environmental problems caused by the use of petrochemicals. Moreover, some adhesives, including formaldehyde, are harmful to health. As a result, bio-based wood adhesives have recently attracted much attention.

Various types of bio-based wood adhesives have been reported, including protein-based [7], oil-based [8], carbohydrate-based [9,10,11], tannin-based [12,13,14], and lignin-based adhesives [15,16,17,18]. Recently, citric acid-based wood adhesive has been investigated as one of the few known chemical bonding-type wood adhesives [19,20]. Generally, wood adhesive mechanisms can be categorized into three types: those involving mechanical interactions, physical interactions, and chemical interactions. Chemical interactions are the strongest because of the covalent bond between adhesive and wood.

Wood biomass, which includes various components such as cellulose, hemicellulose, and lignin, has many reaction sites and should be a reactable adherend. During the process of adhesion with a citric acid-based wood adhesive, esterification is thought to occur between citric acid and the various wood components. Due to the formation of such ester linkages, citric acid wood adhesive provides excellent dimensional stability and water resistance in wood moldings [21]. Citric acid has three carboxyl groups and functions like a staple in the moldings. The changes to the chemical structure after molding have been reported by infrared (IR) analysis [22,23], solid-state cross-polarization magic angle spinning carbon-13 nuclear magnetic resonance (CP-MAS ^13^C NMR) analysis [24,25] and matrix assisted laser desorption/ionization time-of-flight mass spectropy (MALDI-TOF MS) analysis [25]. In the case of IR analysis, the absorbance bands of ester can be detected easily, although the band of the carbonyl group is overlapped with that of the carboxyl group. In CP-MAS ^13^C NMR, the peaks of the C-bonded ester can be obtained directly, although the signals are broad and therefore overlap. In MALDI-TOF MS, parts of bond structures can be predicted from MS data of fragments from wood veneer board powder. Thus, these methods can confirm some data about the bonding between citric acid and wood components, and they have been able to provide some detailed information regarding bonding structures. However, there still remains some uncleared information regarding chemical bonding structures.

Here, we attempt to elucidate the bonding structure between citric acid and wood components with Fourier-transform infrared spectroscopy (FT-IR) and heteronuclear single quantum coherence-nuclear magnetic resonance (HSQC-NMR) analysis. Much of the information about the chemical structure of wood biomass has been obtained by the nuclear magnetic resonance (NMR method) [26,27,28]. Specifically, 2D NMRs, such as HSQC-NMR spectroscopy, are appropriate for the more detailed structural analysis because they allow removal of the overlapping that occurs in 1D NMRs (^1^H and ^13^C), Kim and Ralph recently developed a gel-state 2D NMR (HSQC-NMR) method for wood biomass [29,30,31]. This method can characterize the structures of hemicellulose and lignin in wood biomass without the need to purify each component. In the field of wood adhesives, Yelle et al. reported an HSQC-NMR analysis of wood materials molded with polymeric methylene diphenyl diisocyanate (pMDI), a wood adhesive [32,33]. It appears this method could increase our knowledge about citric acid-based wood adhesive substantially.

In this study, we analyze the bonding structures between citric acid and wood powder with 2D HSQC-NMR spectroscopy to elucidate the reaction mechanism underlying the adhesive process.

## 2. Materials and Methods

### 2.1. Materials

The wood powder was derived from *sugi* (*Cryptomeria Japonica*). Citric acid was purchased from Nacalai Tesque, Inc. (Kyoto, Japan) and ground with a mortar to a powder with a particle size less than 250 μm. These materials were dried in vacuo at 60 °C for 15 h. DMSO-d6 and pyridine-d5 were purchased from Wakenyaku Co., Ltd. (Kyoto, Japan).

### 2.2. Molding Preparation

The wood powder (6.4 g) and citric acid powder (1.6 g) were mixed until uniformly distributed. The citric acid content (wt%) was 20%. A cylindrical mold with a 70-mm inner diameter was used for the molding. The powder mixture produced as described above was poured into the molds and hot-pressed at 180 °C and 4MPa for 10 min. The obtained molding was stored in a desiccator with silica gel.

### 2.3. FT-IR Spectroscopy

FT-IR spectroscopy was performed with a PerkinElmer Spectrum FT-IR spectrometer (PerkinElmer, Boston, MA, USA). The attenuated total reflectance (ATR) technique was applied. The peak areas of the carbonyl groups in the obtained spectra were calculated using PerkinElmer spectrum software after two common preprocessing steps that served as a baseline correction followed by a normalization to the peak at 1315 cm^−1.^

### 2.4. Sample Preparation in HSQC-NMR

The ball-milled NMR samples were prepared according to the method described in the literature [28]. The crushed molding (200 mg) was ball-milled with a planetary Mono Mill PULVERISETTE 6 classic line (Fritsch GmbH, Idar-Oberstein, Germany), with vibration at 600 rpm in zirconium dioxide (ZrO_2_) vessels (12 mL) containing ZrO_2_. 50 balls (5 mm × 5 mm). The grinding time was 80 min, in 20 min grinding/10 min intervals for four cycles to avoid excessive heating. The ball-milled sample was obtained.

### 2.5. 2D HSQC-NMR Spectroscopy

A Bruker Avance 800 MHz spectrometer (Ettlingen, Germany) was used at 323 K. The central DMSO solvent peak was used as the internal standard (δH 2.49, δH 39.5 ppm). An HSQC experiment was applied to the ^1^H−^13^C correlation (Bruker standard pulse sequence “hsqcetgpsp.3”). The solvent was premixed with DMSO-d6/pyridine-d5 (*v*/*v*, 4/1), and samples were swelled in the solvent. The spectral widths were 7788 and 21135 Hz for the ^1^H and ^13^C dimensions, respectively. The number of collected complex points was 1536 for the ^1^H dimension, with a recycle delay of 1.5 or 2.5 s. The number of transients was 16 or 32 and 256 time-increments were always recorded in the ^13^C dimension. Gaussian multiplication (LB = −1.00) and sine-bell squared (SSB = 2) window functions were applied in the ^1^H and ^13^C dimensions, respectively. Prior to the FT, the data matrices were zero-filled up to 2048 and 1024 points in the ^1^H and ^13^C dimensions, respectively.

## 3. Results and Discussion

To clarify the chemical bonds formed between wood and a citric acid-based wood adhesive, we analyzed a wood-based molding, derived from wood powder and citric acid using FT-IR and HSQC-NMR spectroscopy. Figure 1 presents the FT-IR spectra of the wood powder (Figure 1a) and the moldings (Figure 1b,c). The carbonyl peak at 1740 cm^−1^ of the esters formed between citric acid and the hydroxyl groups in wood or carboxylic acids from citric acid appears in the FT-IR spectrum (Figure 1b) [22,34,35]. However, FT-IR analysis cannot separate the only peak of esters from the peak. The only signal of chemical bonding between wood and citric acid in the FT-IR spectrum could not be obtained unless unreacted citric acid was removed from the molding (Figure 1c). Therefore, the molding was analyzed using HSQC-NMR to obtain more detailed information without removing the citric acid.

The HSQC-NMR spectra of the wood powder (*sugi*) and the wood-based molding are shown in Figure 2. In the spectrum of *sugi* (Figure 2a), signals of polysaccharide (cellulose, glucomannan, xylan) and lignin (β-O-4, β-5, β-β substructures) were observed, and assigned by comparison with those in the literature [29,30,36,37]. First, we focused on the lignin substructures. The H_α_/C_α_ signal (4.88/71.5 ppm), H_β_/C_β_ signal (4.41/83.9 ppm (*erythro*), 4.36/84.6 ppm (*threo*)), and H_γ_/C_γ_ signal (approximately 3.40~3.70/60.2 ppm) of the β-O-4 lignin substructure (cyan) were observed, in addition to the H_α_/C_α_ signal (5.55/8.71 ppm), H_β_/C_β_ signal (3.56/54.3 ppm), and H_γ_/C_γ_ signal (approximately 3.75/63.5 ppm) of the β-5 lignin substructure (green). We also detected the H_α_/C_α_ signal (4.66/85.0 ppm), H_β_/C_β_ signal (3.05/54.0 ppm), and H_γ_/C_γ_ signal (3.79, 4.11/71.0 ppm) of the β-β lignin substructure (purple). We next considered the polysaccharide substructures. The H_2_/C_2_ signal (5.41/70.8 ppm) of the 2-acetyl mannose unit (orange) and the H_3_/C_3_ signal (4.94/73.7 ppm) of the 3-acetyl mannose unit (orange) were evident. The other signals of the mannose unit could not be assigned due to overlapping. Finally, the xylan unit signals (blue) and cellulose unit signals (red) could be observed. A portion of the glucose residues in galactoglucomannan may overlap with the cellulose unit signals.

### 3.1. Esterification between Wood Components and Citric Acid

Compared with the aliphatic region (the 45–90/2.0–6.5 ppm) of the *sugi*’s spectrum in Figure 2a, a signal appeared at approximately 4.18/64.5 ppm (red circle) after molding (Figure 2b). This signal was actually two overlapping signals—namely, the esterified 6-position of polysaccharides (such as cellulose, galactoglucomannan) and the H_γ_/C_γ_ signal of the γ-esterified β-O-4 lignin substructure. The H_γ_/C_γ_ signal of the γ-esterified β-5 lignin substructure (green) were also observed at 4.33/65.9 ppm. On the other hand, the H_α_/C_α_ signal of the α-esterified β-O-4 lignin substructure was never detected.

We then focused on the H_β_/C_β_ signals of β-O-4 and β-5 lignin substructures at 4.41/83.9 (*erythro*) and 4.36/84.6 (*threo*) ppm (cyan), and 3.51/53.5 ppm (green), which can be affected by a functional group at the α- or γ-position. Although the original H_β_/C_β_ signals remained, the H_β_/C_β_ signals of γ-esterified β-O-4 and β-5 lignin substructures were found at 4.61/81.0 and 4.72/82.1 ppm (cyan), and 3.60/49.5 ppm (green), in the spectrum of the wood-based molding. This confirmed that esterification of a part of the γ-position in the β-O-4 and β-5 lignin substructures proceeded under the molding condition.

Moreover, one more new signal at 4.67/74.0 ppm (blue) appeared after molding, although its signal volume was small. This signal corresponded with the H_2_/C_2_ signal of the esterified xylan. No other esterified xylan signals were observed. It was found that the reactivities to the secondary and benzyl hydroxyl groups were lower because of the steric hindrance of the citric acid, which is bulky.

In conclusion, primary hydroxyl groups of polysaccharides (cellulose and glucomannan) and lignin substructure (β-O-4 and β-5), and a secondary hydroxyl group at the 2-position of xylan were partly esterified with citric acid under the molding condition employed herein (Figure 3) and the result insisted that citric acid can function as binder like Figure 4. In consideration of these results and the molding condition, the expected reaction mechanism should be Fischer esterification, as shown in Figure 5.

Figure 2c,d are focused on the aromatic region in the 95–150/5.8–8.6 ppm. Although there is almost no change in any signals in these spectra, the G’ signal volume in the spectrum of the wood-based molding is slightly higher than the *sugi’s* spectrum. This result confirmed that the α-positions of lignin were slightly oxidized during the molding process.

### 3.2. Cleavage of Polysaccharide under the Citric Acid Condition

We then turned to the polysaccharide’s anomeric region at 90–110/3.5–6.0 ppm to elucidate the effect of citric acid on polysaccharides. In the anomeric region, the H_1_/C_1_ signals of polysaccharides appeared. H_1_/C_1_ signals of cellulose (β-D-Glc*p*), xylan (β-D-Xyl*p*) and mannan (β-D-Man*p*) were observed at 4.46/102.9, 4.37/101.9 and 4.64/100.5 ppm, respectively. After molding, the H_1_/C_1_ signal volumes of xylan (β-D-Xyl*p*) and mannan (β-D-Man*p*) decreased more than that of cellulose (β-D-Glc*p*). Then, the two forms of reducing ends—the axial −OH (α) and equatorial −OH (β)—appeared in these spectra. In the case of cellulose and xylan, the H_1_/C_1_ signals of the α and β reducing ends were observed in these spectra. The H_1_/C_1_ signals of the reducing ends (α) of cellulose and xylan overlapped at 5.04/92.6 ppm; those of the reducing end (β) of cellulose and xylan were observed at 4.46/97.1 ppm and 4.38/97.9 ppm, respectively. In the only spectrum after molding, the H_1_/C_1_ signal of mannan’s reducing ends (α) appeared at 5.04/94.2 ppm. Compared to the spectra of wood components before molding (Figure 2e) and after molding (Figure 2f), the signal volumes of reducing ends in Figure 2f increased.

These results revealed that some glycoside linkages of cellulose and hemicelluloses, such as galactoglucomannan and (arabino)glucuronoxylan, were cleaved under the molding condition to release the reducing ends. We assume that the formation of xylan’s reducing end caused an increase in the reactivity at the 2-position for esterification. Figure 6 shows the cleavage mechanism we consider likely. In the case of cellulose and xylan, the reducing end formation, produced by cleaving the glycoside bond, depends on the water addition direction—axial or equatorial to produce the α or β reducing ends. On the other hand, in the case of mannan, only one type of reducing end (α) is produced because of the effect of the 2-position of mannose residues.

## 4. Conclusions

From the results of this study, the ester formation between citric acid-type natural wood adhesive and wood components under the adhesive proceeding was directly confirmed and clarified in detail by HSQC-NMR monitoring. In the esterification, the primary hydroxyl groups of polysaccharides and lignin substructures reacted preferentially, and a little part of the secondary hydroxyl group at the 2-position of D-xylosyl. On the other hand, a part of polysaccharides, specifically hemicelluloses, were degraded simultaneously, partly by the cleavage of their glycoside linkages under the citric acid condition, to produce the reducing end under the adhesive proceeding. Citric acid is bonded with wood components and the bonds improve the wood molding properties, although polysaccharides are damaged slightly. We conclude that the citric acid-type adhesive is a chemical covalent bonding-type adhesive and citric acid functions as a clamp between wood powder particles.

## Figures and Tables

**Figure 1 polymers-13-00058-f001:**
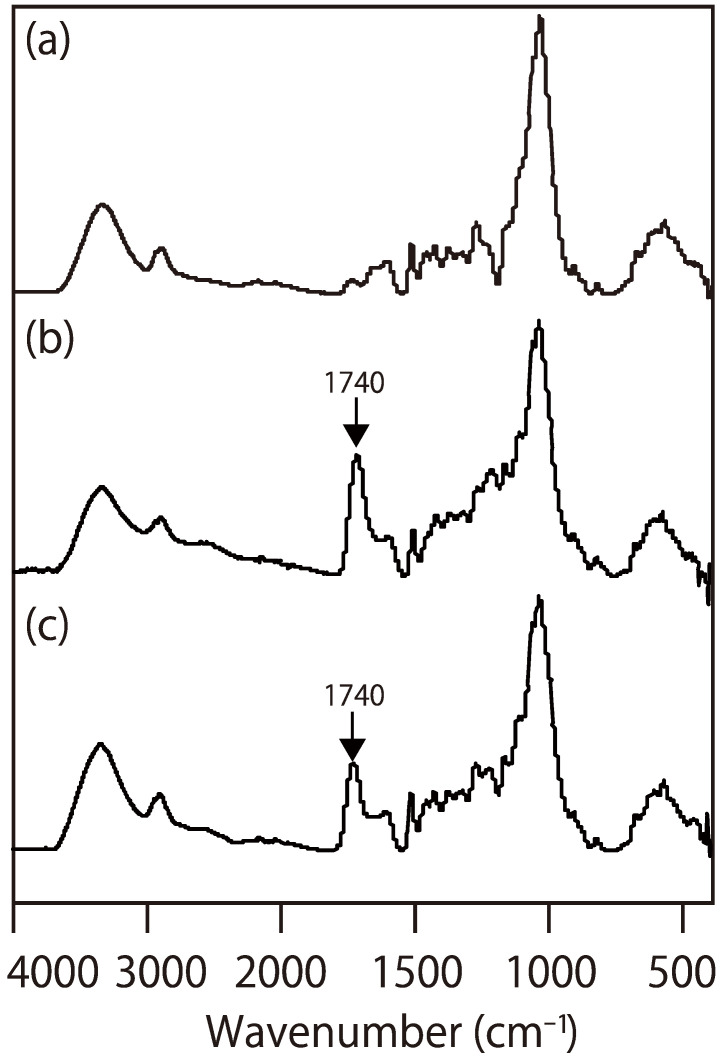
Fourier-transform infrared spectroscopy (FT-IR) spectra of: (**a**) wood (sugi); (**b**) the wood-based molding (W80) before the removal of citric acid; and (**c**) the wood-based molding (W80) after the removal of citric acid.

**Figure 2 polymers-13-00058-f002:**
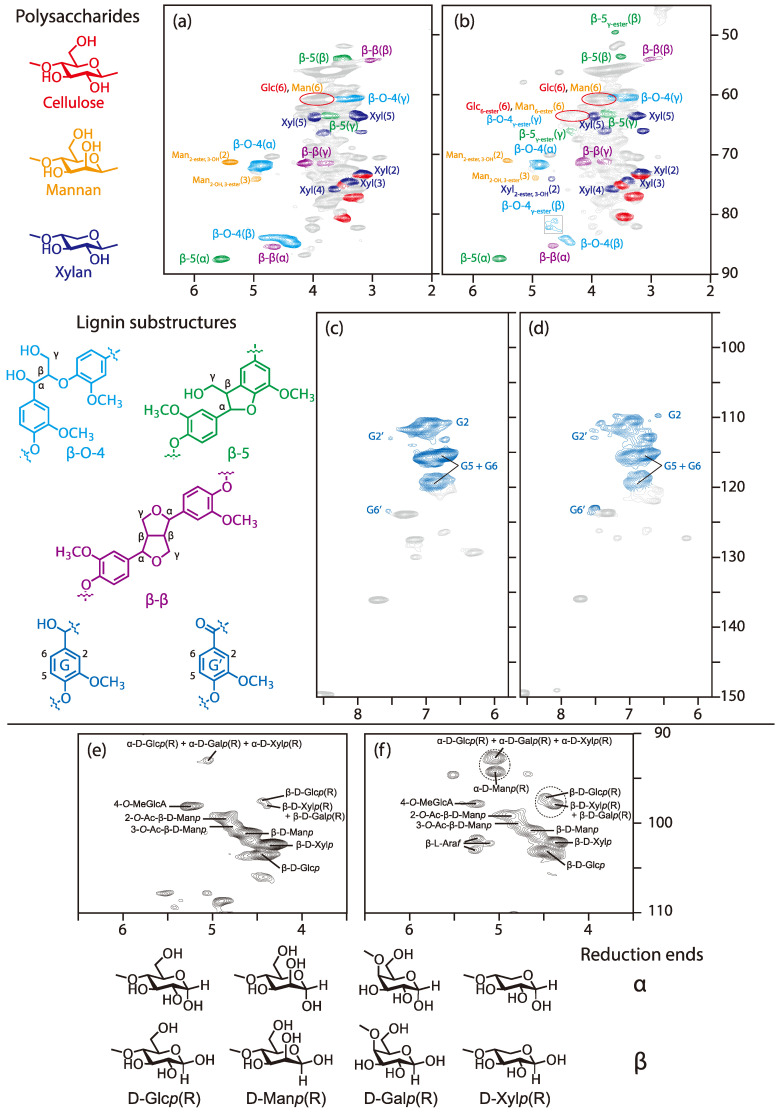
Heteronuclear single quantum coherence-nuclear magnetic resonance (HSQC NMR) spectra of wood (*sugi*) and the wood-based molding (W80): (**a**) *sugi* (aliphatic region); (**b**) W80 (aliphatic region); (**c**) *sugi* (aromatic region); (**d**) W80 (aromatic region); (**e**) *sugi* (anomeric region); and (**f**) W80 (anomeric region).

**Figure 3 polymers-13-00058-f003:**
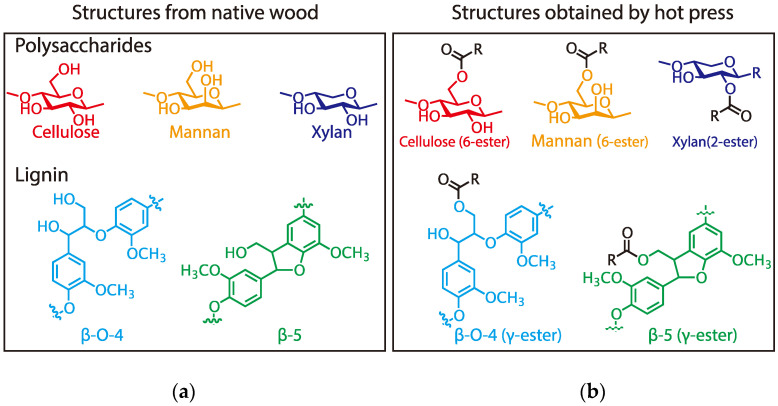
Esterification of wood components with citric acid for (**a**) structures from native wood, and (**b**) structures obtained by hot press.

**Figure 4 polymers-13-00058-f004:**
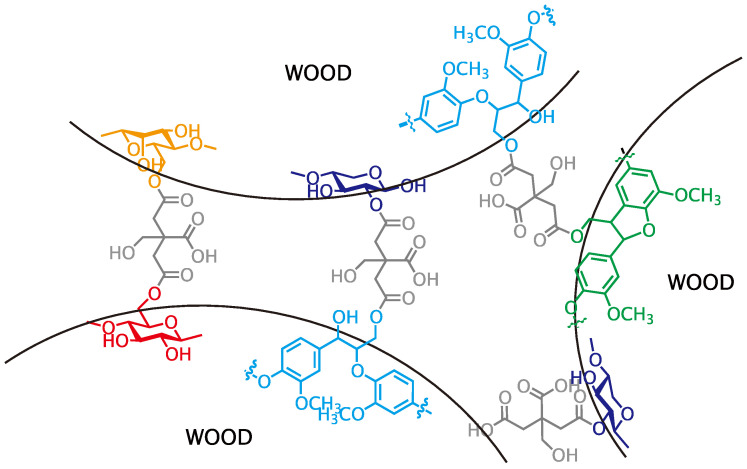
Structure between wood powders.

**Figure 5 polymers-13-00058-f005:**
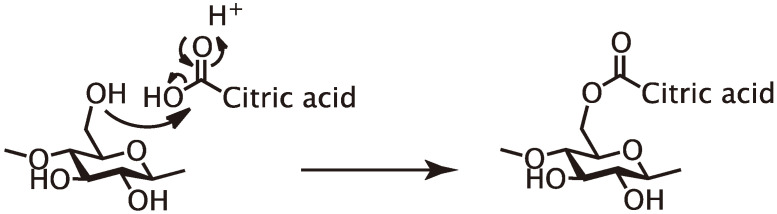
The reaction mechanism under the citric acid-based adhesive process.

**Figure 6 polymers-13-00058-f006:**
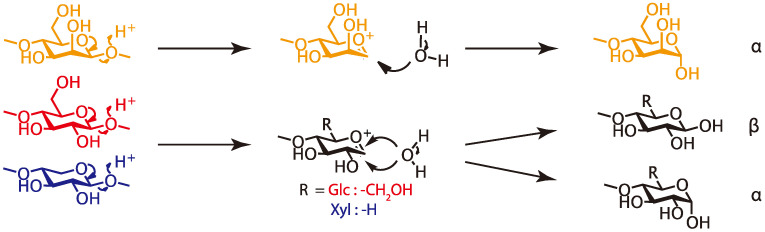
Cleavage of the glycoside linkages.

## Data Availability

The data presented in this study are available on request from the corresponding author.

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
