# Peer review of "Bond Structures between Wood Components and Citric Acid in Wood-Based Molding"

_polymers, 2020, doi:10.3390/polym13010058_

Round 1

Reviewer 1 Report

please see my comments in the attached file

Reviewer 2 Report

This paper refers the chemical bonding structures between citric acid and wood by heteronuclear single quantum coherence-nuclear magnetic resonance (HSQC-NMR) analysis of wood-based molding using Japanese cedar (Cryptomeria japonica) and citric acid. The results showed that the esterification of hemicellulose and lignin, and degradation of hemicellulose, proceeded under the molding condition, which will help inform our understanding of the adhesive mechanism in citric acid-based wood adhesive.

Following is a suggested list of amendments/additions. 

  1. An English revision by a native English-speaking person/agency is a must. There are improvements needed. 

  1. Title– It is better if the title “ Bond between wood components and citric acid in wood-based molding” change to “Bond structure between wood components and citric acid in wood-based molding”.

  1. Materials and Methods Page 2 Line 71– They are indirect methods for the Fourier-transform (FT)-IR and heteronuclear single quantum coherence-nuclear magnetic resonance (HSQC-NMR) analysis, therefore, I suggest that some direct methods should be considered .

  1. Materials and Methods Page 2– How many samples were prepared for Fourier-transform (FT)-IR measurement?

  1. Page 3 Line 108–  It is very hard to find the change of carbonyl peak at 1740cm-1 of esters between citric acid and hydroxyl groups in wood or carboxylic acids from citric acid appears in the FT-IR spectrum from Figure 1. Please mark them in Figure 1 using the arrows.

  1. I think the figure 2 could be modified perfect. For example, more arrowheads should be marked in important locations.

  1. The format in references should be uniform, please revise them.

  1. Relative to the other references, what is your biggest innovation?
